# Towards a Comprehensive Assessment of Statistical versus Soft Computing Models in Hydrology: Application to Monthly Pan Evaporation Prediction

Mohammad Zounemat-Kermani [1], Behrooz Keshtegar [2,*], Ozgur Kisi [3] and Miklas Scholz [4,5,6,7]

1 Department of Water Engineering, Shahid Bahonar University of Kerman, Kerman 7616913439, Iran; zounemat@uk.ac.ir
2 Department of Civil Engineering, University of Zabol, Zabol 9861335856, Iran
3 Department of Civil Engineering, Ilia State University, Tbilisi 0162, Georgia; ozgur.kisi@iliauni.edu.ge
4 Division of Water Resources Engineering, Faculty of Engineering, Lund University, P.O. Box 118, 22100 Lund, Sweden; miklas.scholz@tvrl.lth.se
5 Department of Civil Engineering Science, School of Civil Engineering and the Built Environment, Kingsway Campus, University of Johannesburg, P.O. Box 524, Aukland Park, Johannesburg 2006, South Africa
6 Department of Town Planning, Engineering Networks and Systems, South Ural State University (National Research University), 76, Lenin Prospekt, 454080 Chelyabinsk, Russia
7 Institute of Environmental Engineering, Wroclaw University of Environmental and Life Sciences, ul. Norwida 25, 50-375 Wrocław, Poland
* Correspondence: bkeshtegar@uoz.ac.ir

**Abstract:** This paper evaluates six soft computational models along with three statistical data-driven models for the prediction of pan evaporation (EP). Accordingly, improved kriging—as a novel statistical model—is proposed for accurate predictions of EP for two meteorological stations in Turkey. In the standard kriging model, the input data nonlinearity effects are increased by using a nonlinear map and transferring input data from a polynomial to an exponential basic function. The accuracy, precision, and over/under prediction tendencies of the response surface method, kriging, improved kriging, multilayer perceptron neural network using the Levenberg–Marquardt (MLP-LM) as well as a conjugate gradient (MLP-CG), radial basis function neural network (RBFNN), multivariate adaptive regression spline (MARS), M5Tree and support vector regression (SVR) were compared. Overall, all the applied models were highly capable of predicting monthly EP in both stations with a mean absolute error ($MAE$) < 0.77 mm and a Willmott index ($d$) > 0.95. Considering periodicity as an input parameter, the MLP-LM provided better results than the other methods among the soft computing models ($MAE$ = 0.492 mm and $d$ = 0.981). However, the improved kriging method surpassed all the other models based on the statistical measures ($MAE$ = 0.471 mm and $d$ = 0.983). Finally, the outcomes of the Mann–Whitney test indicated that the applied soft computational models do not have significant superiority over the statistical ones ($p$-value > 0.65 at $\alpha$ = 0.01 and $\alpha$ = 0.05).

**Keywords:** pan evaporation; machine learning models; improved kriging; SVR; MARS

## 1. Introduction

One of the key elements of water resources management and hydrological projects is to estimate the evaporation in a given region. This is even more important in managing water resources in arid and semi-arid regions [1]. Some researchers have applied the Budyko framework, which is a straightforward model that considers only rainfall and potential evaporation as the required input for simulating and controlling various water management plans [2,3]. Simply, knowing the accurate amount of the evaporation is essential for water resources management projects.

Researchers have applied different approaches for modeling pan evaporation (EP) and evapotranspiration in the literature classified as (i) physically-based combination

models that take into account mass and energy conservation principles; (ii) semi-physical models that use either mass or energy conservation; and (iii) data-driven models including soft computing and statistical techniques [4–6]. The shortage of EP data (temporally or spatially) is a major problem in some areas because it is difficult and expensive to install evaporation pans. In these cases, applying data-driven and soft computing models for estimating water evaporation is an effective and appropriate approach [7–9]. The accuracy of modeling approaches is the most important parameter to take into account.

Several researchers have used climatic variables to estimate EP values [10,11]. Climate-based approaches are appropriate methods when provided with specific climatic data, which cannot always be easily obtained from a determined area. Similarly, data-driven approaches, including computational intelligence and machine learning are also suitable for estimating the EP. Recently, regular hybrid and integrative data-driven models (e.g., artificial neural networks (ANN) as well as support vector machine (SVM) and adaptive neuro-fuzzy inference systems (ANFIS)) have been used for estimating the EP [7,12–20].

Tabari et al. [21] estimated the daily EP of a region by using different methods (ANNs and multivariate nonlinear regression (MNLR)) and concluded that ANN was more accurate than MNLR. Kişi et al. [22] applied three soft computing models, such as M5tree, ANN, and chi-squared automatic interaction, to predict the daily EPs in Turkey. They reported that the ANN model performed better than the two others. Tezel and Buyukyildiz [23] investigated the usability of ANNs and $\varepsilon$-SVR to estimate monthly EP. According to the performance criteria, the ANN algorithms and $\varepsilon$-SVR had the same performance. Tezel and Buyukyildiz [23] compared the accuracy of SVM basis $\varepsilon$-support vector regression, radial basis function (RBFNN), and multilayer perceptron ANN (MLPNN), and showed that the latter provided the most accurate results. Keshtegar and Kisi [24] proposed the modified response surface method (RSM) and modified RSMs have been compared with ANFIS and M5Tree. Wang et al. [25] investigated the capabilities of ANFIS, M5Tree, and fuzzy genetic (FG) for six stations in the Yangtze River Basin. The results stated that the FG model generally produced better results. In another study, Wang et al. [26] compared the abilities of FG, SVR, MARS, M5Tree, and multiple linear regression (MLR). The overall results indicated that the soft computing models generally performed better than the regression methods. Ghorbani et al. [27] applied a hybrid MLPNN for daily EP prediction at two stations. The results showed that the MLPNN model provided better performance compared to the SVM model. Majhi et al. [28] applied a deep ANN model and compared it with the traditional MLPNN for three areas of the Chhattisgarh State in India. The findings of the study showed that the deep ANN model was more accurate than the traditional MLPNN. The abilities of ANN and extreme learning machine (ELM) models have been compared in predicting EP for two stations in Algeria by Sebbar et al. [29]. The results indicated that the ELM could be successfully used to estimate the daily EP [29]. Al-Mukhtar [30] investigated the applicability of quantile regression forest for EP prediction in arid areas. In comparison to conventional NNs and linear regression models, the applied quantile regression forest provided better results. Mohammadi et al. [31] predicted monthly EP using integrative ANFIS, MLP and RBNN models for two stations in Iran. The results showed that the integrative ANFIS model acted better compared to the MLP and RBNN. Yaseen et al. [32] applied several machine learning models, including ANN, classification and regression tree (CART), gene expression programming and SVM for predicting EP in arid and semi-arid areas. The findings of the study indicated that the SVM was superior to the other applied models.

A literature review related to the kriging approach revealed that it has never been used to predict PE. However, this method was applied for the prediction of solar radiation [33] and the daily total dissolved gas in aquatic systems concentration by Heddam et al. [34]. The kriging interpolation, which is a flexible regression tool for approximating any nonlinear problem, can be introduced as a potential method for providing accurate EP predictions.

Indeed, soft computing techniques have provided satisfactory results in EP prediction [35]. The majority of EP modeling reported the superiority of soft computing models

over statistical models [21,26,36]. The main objective of this study is to challenge the capability of soft computing techniques versus statistical data-driven models.

The present study investigates the accuracy of six soft computing methods, including M5tree, MARS, SVR, RBFNN, Levenberg–Marquardt perceptron ANN and conjugate gradient perceptron ANN and compares them with three different statistical approaches, RSM, kriging, and improved kriging. In improved kriging, the basic functions are transferred from polynomial to exponential functions to estimate monthly EP. In kriging models, the second-order polynomial function is applied as regressed function to predict nonlinear challenges. This function may not have predicted an accurate result for complex problems such as EP. Thus, novel improved modeling is parented to enhance the regressed function applied in original kriging. It uses a nonlinear transformation as an exponential map for input variables of the EP predictions as a complex engineering problem with nonlinear effect that the accurate results in the modeling process is a cap for prediction of the statistical regression approaches such as kriging and RSM models.

To our best knowledge, similar studies have not been carried out in applying the above-mentioned methods for the estimation of EP. The subsequent parts of the rest of the paper are organized as follows: In Section 2, the two stations are introduced, and the data sets are presented. The third section deals with nine modeling methods applied in statistical and soft computing approaches. The results of the predicted EP are presented and compared in the fourth section. The fifth section of the paper deals with the hypothesis testing and relevant discussion. Finally, the last section provides the concluding remarks of the present work.

## 2. Case Study and Dataset

The input parameters for this study are monthly climatic data such as solar radiation (SR, as Langley), sunshine hours (HS), relative humidity (RH), wind speed (WS as m/s) as well as the minimum (Tmin as °C) and maximum temperature (Tmax as °C). Two stations in the Eastern Mediterranean Region as Adana (latitude 37.22° N, longitude 35.40° E and altitude 20 m) and Antakya (latitude 36.33° N, longitude 36.30° E and altitude 100 m) were selected for the comparing modeling results. The map of the study area is illustrated in Figure 1. The studied area has a climate with cool and rainy winters, and moderately hot and dry summers and it receives yearly rainfall amounts of between 580 and 1300 mm. Data were gathered from the Turkish State Meteorological Service (TSMS) having a modernized calibration center. The calibration center was accredited by the Turkish Accreditation Agency to ensure the measurements' reliability and to provide the validity of the measurements' quality around the world. Temperature, RH, and WS calibration laboratories are accredited with TS EN ISO/IEC 17025 standards and work in accordance with this standard. Global radiation and wind direction data are also in accordance with the TS EN ISO/IEC 17025 standards. The evaporimeter used for obtaining pan evaporation in Turkey is the US Weather Bureau Class A pan. The raw datasets were directly utilized in the presented study without pre-processing. The available data period covers September 1981 to March 2016 for Adana and from October 1983 to December 2010 for Antakya. There is no gap in the data.

In Figure 2, the general characteristics of the independent variables and the target value for the (a) Adana and (b) Antakya stations are shown using the box-whisker plot and related correlation values. These plots graphically depict the variability of each parameter in terms of minimum, quartiles, and maximum values. Moreover, outliers are plotted as individual points.

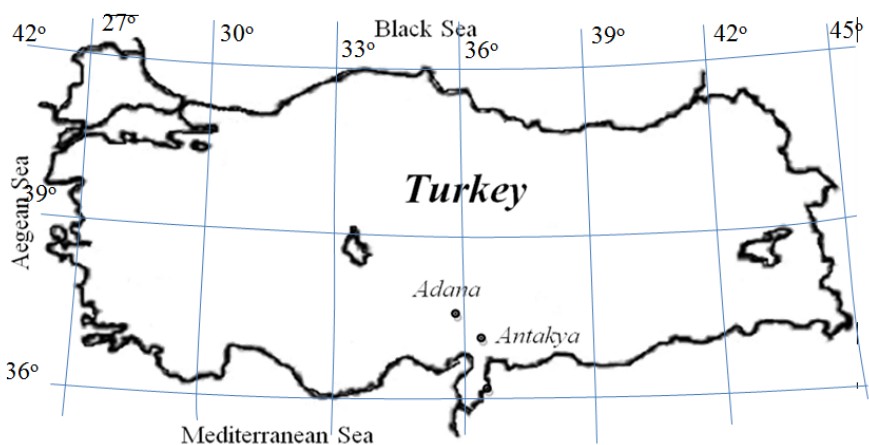

**Figure 1.** The studied stations in the Mediterranean region of Turkey; Adana and Antakya.

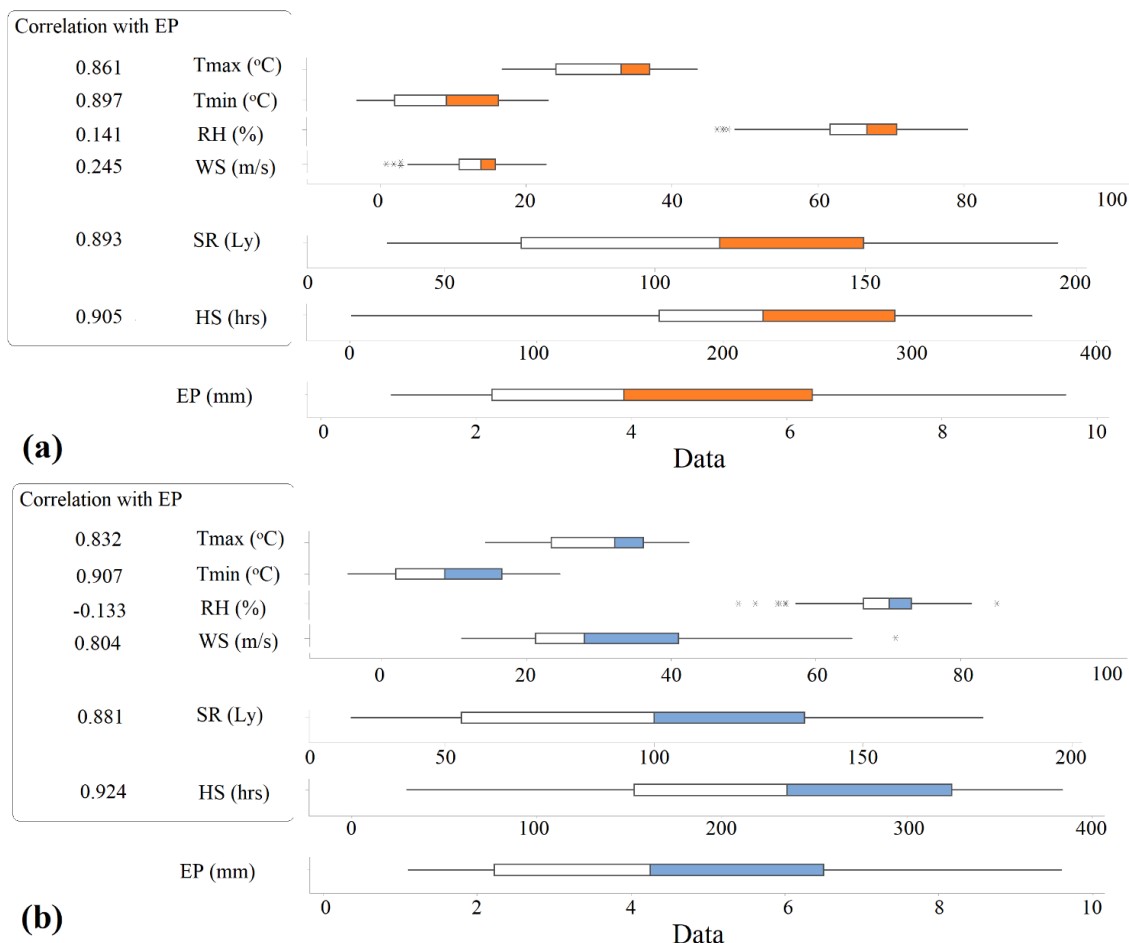

**Figure 2.** Box-plots of independent variables and EP for (**a**) Adana (**b**) Antakya Stations.

The Pearson correlation coefficient was applied to analyze the effects of Tmax, RH, WS, SR, and HS on EP. It can be seen from Figure 2 that there were high correlations between EP, Tmin, Tmax, SR, and HS for both Stations. It is worth noting that the wind speed showed high correlation with EP for the Antakya Station (correlation = 0.804), whereas the reciprocal value of the WS correlation for the Adana Station is much lower (correlation = 0.245).

For both of the Stations, the correlation between the relative humidity (RH) and the EP was weaker than the other parameters. However, in the Antakya Station, the correlation

value of RH is negative, which implies that the increase in relative humidity might lead to a decrease in EP. The main factors responsible for the EP in both Stations include sunshine hours (HS), Tmin, and SR. In the present study, data was split into two sets, 70% (for training) and 30% (for testing) for executing and assessing the applied models.

As for the sensitivity analysis, the best subset regression using adjusted R-Sq and Mallows CP was applied. The results indicated that all of the input variables have a significant impact on the EP variable; hence, all of the independent parameters are used as the input vector for constructing the models. Considering a descending order (from the most influential to the least influential independent variables on PE), the following results were achieved:

- Antakya Station: HS, Tmin, SR, WS, Tmax, and RH.
- Adana Station: HS, Tmin, SR, Tmax, WS, and RH.

### 3. Methods

Nine different approaches in terms of two main categories of statistical (RSM, kriging and improved kriging) and machine learning models (SVR, MARS, M5Tree, MLP-LM, MLP-CG and RBNN) were implemented for estimating EP.

### 3.1. Artificial Neural Networks: MLP-LM, MLP-CG, RBFNN

The ANNs are adaptable learning structures constructed from different interconnected layers and a number of processing elements (called artificial neurons). So far, several types of ANNs were developed and implemented for simulating and predicting hydrological problems such as evaporation [37]. Among all the developed models, the multilayer perceptron (MLP) and radial basis function neural network (RBFNN) have been used in several applications, and their potential in capturing nonlinear features of complex phenomena were proven by the following relation [38,39].

$$\hat{Y}(x) = [\beta_0 + \sum_{j=1}^{M} w_j f(\beta_j + \sum_{i=1}^{NV} w_{ij} x_i)] \tag{1}$$

where $\beta_0$, $\beta_j$, $w_j$, $w_{ij}$ are respectively the biases and weights of the output and the M-hidden layer and NV represents the number of input variables. $f$ is an activation function for hidden neurons in the MLP and RBFNN models. Sigmoid functions were considered for the MLP and radial basis function were applied for the RBFNN models.

It should be noted that MLPs [38] and RBFNNs [40] can be considered as the fundamental versions of feed-forward networks with a supervised learning approach. In this study, two types of MLP networks have been developed using two different approaches for the learning algorithm: (1) the Levenberg–Marquardt algorithm, and (2) the conjugate gradient (CG) algorithm. In addition to the MLP-LM and MLP-CG neural networks, the efficiency of RBFNN was also challenged for the evaporation simulation [41].

### 3.2. Support Vector Regression (SVR)

The rapid application of SVMs in modeling various problems in engineering urges researchers to apply different types of SVMs to different research fields. The core analogy of constructing SVMs is to map variables from input space into high-dimensional feature space by using special functions as below [42,43]:

$$\hat{Y}(x) = \beta_0 + \sum_{i=1}^{N} (\alpha_i - \alpha_i^*) K(x, x_i) \tag{2}$$

where $\beta_0$ is bias and $K(x, x_i)$ is the Kernel function for transferring the input data from x-space to N-set feature space which is computed as below relation [44]:

$$K(x, x_i) = \exp(-\frac{\| x - x_i \|^2}{2\sigma^2}) \tag{3}$$

where, $\sigma$ is the parameter of the Kernel function. $\alpha_i$ and $\alpha_i^*$ represent the Lagrange multipliers as unknown coefficients in the SVR model. Recently, the application of the SVR model in hydrological time series modeling has provided promising outcomes [45]. Several researchers have already claimed that SVR is efficient in modeling evaporation processes [23,46].

### 3.3. Multivariate Regression Spline (MARS)

Proposed by Friedman in 1991 [47], multivariate adaptive regression spline is a procedure for fitting adaptive nonlinear functions using a piecewise nonparametric regression method. Unlike the black box models (e.g., ANNs), MARS models are deterministic, which means that in the final regression form the input variables are identified and the interactions between them are specified. Therefore, the MARS models are much easier to be interpreted than the other techniques [48–50]. Considering X as the only independent variable and Y as the dependent variable (target value), it can be seen in Figure 3 that the space of X variable is divided into three sub-regions with three different equations. These equations relate the independent variable space to the target of the system.

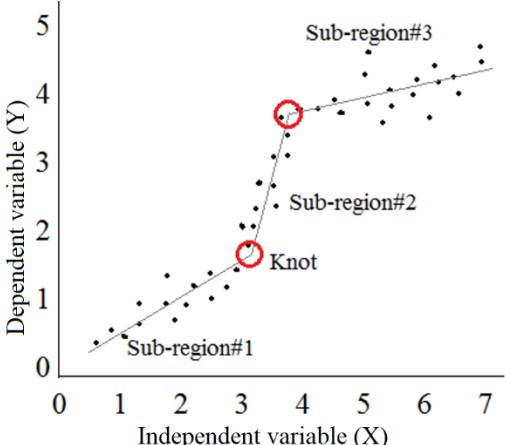

**Figure 3.** A schematic sketch for the illustration of sub-regions of the MARS method.

The endpoints of the segments of each sub-region are called knots (Figure 3). The resulting piecewise regression lines (basis functions, *BF*s) make the final regression form flexible and appropriate for capturing trends from linear functions as bellow [51]:

$$\hat{Y}(x) = \beta_0 + \sum_{i=1}^{m} \beta_i BF_i \tag{4}$$

where $\beta_i$ = 0, 1, ... , $m$ are unknown coefficients and $m$ is the number of basis functions (*BF*) which is determined using a piecewise linear function as follows [33]:

$$BFi = \{\max(0, x - C_i), \ \max(0, C_i - x)\} \tag{5}$$

where $C$ represents the knot which is a constant coefficient. By considering more independent variables, more equations will be added to the final regression form. In order to determine the location of knots, an adaptive regression algorithm is used. In addition, BFs are generated by a stepwise searching process. In brief, the main procedure of the MARS method is categorized into two parts of the forward and backward phases. The location

of potential knots and BFs equations are specified in the forward phase. To modify and improve the modeling accuracy, unnecessary and the least effective variables are removed in the backward phase [48]. Further details for the mathematical procedure of the MARS method can be obtained from Friedman [47].

### 3.4. M5 Model Tree

Quinlan (1992) introduced a piecewise linear regression model, called the M5 model tree (M5Tree) [52,53], which has a tree structure based on binary decisions. The linear regression functions, which develop interconnection between the input and output vectors, can be extracted at the terminals (leaves) nodes.

Constructing an M5tree model requires two distinct phases; first the initial tree is generated and is then pruned. In the first phase, data sets are split into several subsets, which create a decision tree. In other words, the M5 model tree splits the data set space into subsets (sub-spaces) and generates a linear regression model [54,55]. As can be observed in Figure 4, the two-dimensional dynamic space of the input vector (X1 & X2) is split schematically into five subsets.

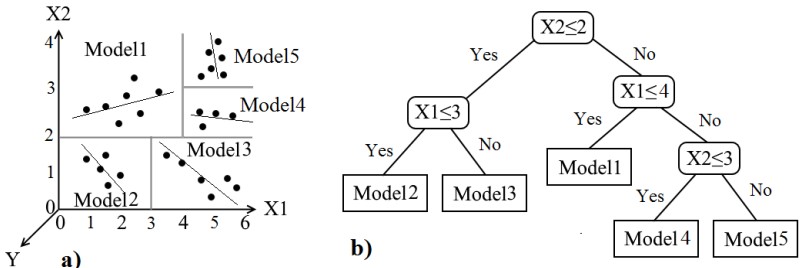

**Figure 4.** Basic sketch for the M5 tree model; (**a**) splitting the input vector into subsets, (**b**) M5 tree structure.

The splitting criterion is determined by assuming the standard deviation (sd) of the class values that reach a node. Based on the sd, the standard deviation reduction (SDR) can be calculated as the following relation [13,56]:

$$SDR = sd(T) - \sum_{i=1}^{n} \frac{|T_i|}{|T|} sd(T_i) \tag{6}$$

where $T$ stands for the set of examples that reaches the node, and $T_i$ is the subset of examples with the $i$th outcome of the potential set. After the first phase (viz. constructing the initial tree), a huge tree-like structure might be generated, which may cause poor generalization. To cope with this problem, in the second phase, the overgrown tree is pruned.

### 3.5. Response Surface Methodology

The response surface methodology (RSM) presents the advantage of multiple regression analysis via a statistical technique to simulate a response space based on quantitative data obtained from the extracted multivariate equations as presented below [57]:

$$\hat{Y}(x) = \beta_0 + \sum_{i=1}^{NV} \beta_i x_i + \sum_{i=1}^{NV} \sum_{j=i}^{NV} \beta_{ij} x_i x_j \tag{7}$$

where $NV$ denotes the number of input variables. $\beta_0$, $\beta_i$ and $\beta_{ij}$ are unknown coefficients for polynomial terms. During the mathematical process, RSM explores the influence of multiple independent variables on the response parameter and optimizes the trending procedure by tuning the number of required experiments [58–61].

### 3.6. Kriging Interpolation Approach

The kriging basis nonlinear model is a well-known interpolation approach to approximate geological problems [62]. It is defined by using stochastic terms according to the following relation [63,64]:

$$\hat{Y}(\boldsymbol{x}) = \hat{\beta}f + r^T(X)R^{-1}(Y - \hat{\beta}f) \tag{8}$$

where $\hat{\beta} = [\hat{\beta}_1, \hat{\beta}_2, \ldots, \hat{\beta}_m]^T$ are regression coefficients for n-support points with m basic functions. The unknown coefficients are computed as follows [65]:

$$\hat{\beta} = \left(f^T R^{-1} f\right)^{-1} f^T R^{-1} Y \tag{9}$$

where $\hat{Y}(X)$ is the predicted value. $R$ represents the correlation matrix which is given as:

$$R = \begin{bmatrix} 1 & r(X_1, X_2) & \cdots & r(X_1, X_n) \\ r(X_2, X_1) & 1 & & r(X_2, X_n) \\ & \vdots & \ddots & \vdots \\ r(X_n, X_1) & r(X_n, X_1) & \cdots & 1 \end{bmatrix} \tag{10}$$

in which $r(X_i, X_j)$ is the cross-correlation function computed by the following relation:

$$r(X_i, X_j) = e^{\theta r_{ij}^2}, r_{ij} = \| X_i - X_j \| \tag{11}$$

where, $r_{ij}$ is the distance between points, $X_i$ and $X_j$ and $\theta > 0$ are unknown correlation parameters, which are determined as presented below [66–69]:

$$\theta = argMax(\frac{log[detR] + nlog[\hat{\sigma}^2]}{2}) \tag{12}$$

where $n$ represents the number of training points, and $\hat{\sigma}^2$ denotes the variance of the model obtained as:

$$\hat{\sigma}^2 = \frac{(Y - \hat{\beta}f)^T R^{-1} (Y - \hat{\beta}f)}{n} \tag{13}$$

In the kriging model, the basis function $f$ can be defined as below:

$$f = \begin{bmatrix} f_1(X_1) & f_2(X_1) & \cdots & f_m(X_1) \\ f_1(X_2) & f_2(X_2) & & f_m(X_2) \\ & \vdots & \ddots & \vdots \\ f_1(X_n) & f_2(X_n) & \cdots & f_m(X_n) \end{bmatrix} \tag{14}$$

where the vector $[f_1(X_1), f_2(X_1), \ldots, f_m(X_1)]$ includes the basic functions that are evaluated at the data input point of $X_1$, and $m$ is the number of the basic functions. The basis function $f$ can be used as polynomial and exponential functions for original kriging and improved kriging, as presented in this study.

In the kriging models, the basic functions are considered as follows:

$$f(X_k) = \{1X_k\} \, where \, X = Mon, Ws, T_{max}, T_{min}, RH, SR, Hs \tag{15}$$

where *Mon* represents the periodicity (month of the year), *Ws* is wind speed (m/s), $T_{max}$ and $T_{min}$ are respectively the maximum and minimum temperature (°C), *RH* is the relative humidity (%), *SR* is the solar radiation (langley), and *Hs* represents the hours of sunshine (h). The surrogate model that uses an adaptive kriging framework can be used for (i) reducing the computational burden and increase the accuracy results of the optimization

problem [50,70], (ii) structural reliability analysis [65,68], and (iii) reliability-based design optimization [67,71].

### 3.7. Improved Kriging

In the fitness process of the kriging model, the basic function term, i.e., $\hat{\beta}f$ is an important factor for providing a flexible prediction. The stochastic term, i.e., $r^T(X)R^{-1}(Y - \hat{\beta}f)$, may produce a smaller covariance for approximating data with accurate basic function. Thus, the nonlinear form of the basic function can improve the accuracy of the EP predictions. A schematic comparative view of the exponential and linear polynomial functions is presented in Figure 5 to illustrate the fitness of the exponential basic function. We used the exponential basic function for the regression process instead of the linear basic function, in order to enhance the ability of the standard kriging model.

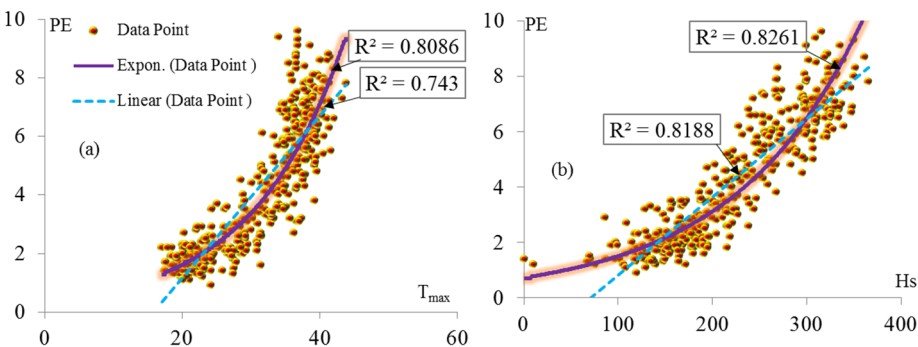

**Figure 5.** Schematic view of basic function using linear and exponential forms for data of (**a**) maximum temperature and (**b**) hours of sunshine.

In improved kriging, the linear and exponential functions are used by the following basic function:

$$f(X_k) = \{1 X_k exp(X_k)\} \tag{16}$$

where $X_k$ are the input variables and exp denotes the exponential operator. The prediction accuracy of the improved kriging model for the estimation of the EP is tested based on an untried data set using $r(X)$ in Equation (11) and predicted relation in Equation (8).

### 3.8. Methodology and Models Evaluation

The modeling process focuses on the monthly predictions of the EP based on two different scenarios, as presented below:

- Scenario I (without periodicity):

In the first scenario (#I), the monthly averaged of six meteorological parameters including wind speed (WS, m/s), relative humidity (RH, %), solar radiation (SR, Langley), sunshine hours (HS, h), minimum (Tmin, °C), and maximum temperatures (Tmax, °C), are considered as the input vectors of the applied models.

- Scenario II, (with periodicity):

In the second strategy, all of the mentioned independent meteorological parameters along with the time factor formed the input vector.

Due to the fact that in the second scenario, the order of the data is important for modeling, the time series cross-validation technique was applied. Thus, in both of the scenarios, 70% of the data was used for training the models, and the other 30% was used for the testing set. In the current work, the root mean square error (*RMSE*) was used as a measure of accuracy, while the mean bias error (*MBE*) was used as a measure of tendency.

The absolute residual of the standard deviation between the actual EP values and the modeled ones (*RSTD*) was used as a measure of precision as presented below [60,72,73]:

$$RMSE = \sqrt{\frac{1}{N}\sum_{i=1}^{N}(EPm_i - EPo_i)^2} \tag{17}$$

$$MBE = \frac{1}{N}\sum_{i=1}^{N}(EPm_i - EPo_i) \tag{18}$$

$$R_{STD} = |STD_m - STD_O| \tag{19}$$

where $N$ is the number of data, $EPm_i$ represents the modeled EP for the $i$th data and $EPo_i$ stands for the observed EP values for the $i$th data. In addition to the above-mentioned measures, other statistics and criteria such as the mean absolute error (*MAE*), mean absolute percentage error (*MAPE*), Willmott index (*d*), total pan evaporation (Tot-EP), maximum value of the relative error between the calculated and observed EP (MAX (RE)) were also used for the evaluation of the applied methods [58].

$$MAE = \frac{\sum_{i=1}^{N}|EPm_i - EPo_i|}{N} \tag{20}$$

$$MAPE = \frac{1}{N}\sum_{i=1}^{N}\frac{|EPm_i - EPo_i|}{EPo_i} \tag{21}$$

$$d = 1 - \frac{\sum_{i=1}^{N}(EPm_i - EPo_i)^2}{\sum_{i=1}^{N}(|EPm_i - EPmean| + |EPm_i - EPmean|)^2} \tag{22}$$

where *EPmean* is the mean of monthly EP. In this study, the Wilcoxon nonparametric statistical hypothesis test is also implemented to evaluate the performance of statistical versus soft computing models at the 95% confidence level. The maximum relative absolute error ((Max (RE)) was computed as max ($RE_i$) and $RE_i = (|\hat{Y}_i - Y_i|)/Y_i$, where $\hat{Y}_i$ an $Y_i$ indicate the estimated and observed pan evaporation.

## 4. Comparison and Results

### 4.1. Evaluation of the Applied Models

Tables 1 and 2 report the comparison statistics of the applied data-driven models for the Adana station for the first and second scenarios. For the first scenario (without periodicity), the improved kriging model has the lowest *MAE* (0.659 mm), *MAPE* (0.189), *RMSE* (0.843 mm) and the highest *d* (0.964), followed by the SVR model. Based on the *MAE*, *d*, and *RMSE* values, the ANN-CG and RSM models provided the weakest results. However, the M5Tree model gave the worst Max (RE) value (135.32). The mean and total pan evaporations were also better approximated by the improved kriging compared to the other models. In the second scenario, the improved kriging model presented a better value of *MAE* than the SVR model (improved kriging = 0.646 mm vs. SVR = 0.648 mm), but worse values for *RMSE* (improved kriging = 0.821 mm vs. SVR = 0.796 mm) and could not be seen as the being better than the SVR model.

**Table 1.** Comparing the results of the applied models without periodicity (Scenario #1) for Adana station in the testing period.

| Category | Model | *MAE* (mm) | *RMSE* (mm) | *MBE* (mm) | *d* | Max (RE) | Mean * (mm) | STD * (mm) | Tot-EP * (mm) | *MAPE* |
|---|---|---|---|---|---|---|---|---|---|---|
| **Statistical** | kriging | 0.712 | 0.891 | 0.228 | 0.958 | 86.31 | 4.36 | 2.19 | 501.63 | 0.213 |
| | Improved kriging | 0.659 | 0.843 | 0.172 | 0.964 | 72.89 | 4.31 | 2.26 | 495.13 | 0.184 |
| | RSM | 0.736 | 0.933 | 0.142 | 0.956 | 79.08 | 4.28 | 2.29 | 491.71 | 0.205 |
| **Soft computing models** | MARS | 0.701 | 0.855 | 0.221 | 0.962 | 87.78 | 4.35 | 2.23 | 500.78 | 0.203 |
| | M5Tree | 0.704 | 0.890 | 0.197 | 0.960 | 80.77 | 4.33 | 2.29 | 498.11 | 0.190 |
| | SVR | 0.668 | 0.828 | 0.223 | 0.964 | 135.32 | 4.36 | 2.15 | 501.08 | 0.208 |
| | ANN(LM) | 0.685 | 0.861 | 0.197 | 0.962 | 83.32 | 4.33 | 2.25 | 498.01 | 0.195 |
| | ANN(CG) | 0.739 | 0.930 | 0.157 | 0.955 | 66.74 | 4.29 | 2.22 | 493.49 | 0.207 |
| | RBFNN | 0.712 | 0.892 | 0.229 | 0.958 | 86.31 | 4.36 | 2.19 | 501.68 | 0.213 |

* The mean, standard deviation (STD) and total pan evaporation (Tot-EP) of the actual data points are mean = 4.134 mm, STD = 2.256 mm and Tot-EP = 475.4 mm, respectively. Optimal structure of SVR: (C = 10, ε = 0.5, σ = 85), ANN (LM): 6-7-1, ANN (CG): 6-8-1, RBFNN: 6-40-1 (σ = 2).

**Table 2.** Comparing the results of the applied models with periodicity (Scenario #2) for Adana station in the testing period.

| Category | Structures | *MAE* (mm) | *RMSE* (mm) | *MBE* (mm) | *d* | Max (RE) | Mean * (mm) | STD * (mm) | Tot-EP * (mm) | *MAPE* |
|---|---|---|---|---|---|---|---|---|---|---|
| **Statistical** | kriging | 0.730 | 0.912 | 0.224 | 0.957 | 88.79 | 4.36 | 2.20 | 501.11 | 0.220 |
| | Improved kriging | 0.646 | 0.821 | 0.168 | 0.966 | 76.97 | 4.30 | 2.26 | 494.75 | 0.181 |
| | RSM | 0.768 | 0.972 | 0.138 | 0.953 | 100.96 | 4.27 | 2.31 | 491.22 | 0.210 |
| **Soft computing models** | MARS | 0.697 | 0.859 | 0.146 | 0.962 | 63.13 | 4.28 | 2.24 | 492.19 | 0.193 |
| | M5Tree | 0.715 | 0.897 | 0.192 | 0.959 | 80.77 | 4.33 | 2.28 | 497.45 | 0.197 |
| | SVR | 0.648 | 0.796 | 0.173 | 0.966 | 106.42 | 4.31 | 2.12 | 495.30 | 0.200 |
| | MLP-LM | 0.746 | 0.949 | 0.206 | 0.954 | 99.85 | 4.34 | 2.26 | 499.06 | 0.205 |
| | MLP-CG | 0.764 | 0.976 | 0.189 | 0.952 | 71.22 | 4.32 | 2.29 | 497.16 | 0.208 |
| | RBNN | 0.682 | 0.842 | 0.233 | 0.963 | 85.82 | 4.37 | 2.21 | 502.21 | 0.214 |

* The mean, standard deviation (STD) and total pan evaporation (Tot-EP) of the actual test data points are mean = 4.134 mm, STD = 2.256 mm and Tot-EP = 475.4 mm, respectively. Optimal structure of SVR: (C = 5, ε = 0.3, σ = 80), ANN (LM): 7-14-1, ANN (CG): 7-12-1, RBFNN: 7-35-1 (σ = 5).

In general, all of the applied statistical and soft computing models approximated the EP values satisfactory (with *d* > 0.95 and *RMSE* < 1 mm). In Table 3, the best predictive model is presented as the improved kriging model based on attaining two of the highest position of the three elements of accuracy, precision, and tendency.

**Table 3.** The general performance of the applied models in terms of accuracy, precision, and tendency for Adana Station in the testing period.

| Category | Model | Scenario I, without Periodicity | | | | Scenario II, with Periodicity | | | |
|---|---|---|---|---|---|---|---|---|---|
| | | Accuracy | Precision | Tendency | Best Model(s) | Accuracy | Precision | Tendency | Best Model(s) |
| **Statistical** | Kriging | M | L | + | | M | L | + | |
| | Improved kriging | H | H | + | * | H | H | + | * |
| | RSM | L | M | + | | L | L | + | |
| **Soft computing models** | MARS | H | H | + | * | M | H | + | |
| | M5Tree | M | M | + | | M | M | + | |
| | SVR | H | L | + | | H | L | + | |
| | MLP-LM | M | H | + | | L | H | + | |
| | MLP-CG | L | M | + | | L | M | + | |
| | RBNN | L | L | + | | H | M | + | |

Note: Accuracy is based on *RMSE* (mm), precision is based on RSTD, and tendency is based on *MBE*. Accuracy and precision: H: high (3 best values), M: moderate (3 median values), L: low (3 worst values); Tendency: +: Over-predicted (positive values); N: neutral (absolute value < 0.01 mm). Best models have been chosen based on acted best at least in two of the three criteria.

Figure 6 demonstrates the observed and estimated EP values of the applied models for the two scenarios, the Adana Station, (a) without periodicity, and (b) with periodicity. It is clear from the fit line equations and $R^2$ values that the improved kriging model has less distributed properties than the other models for both cases.

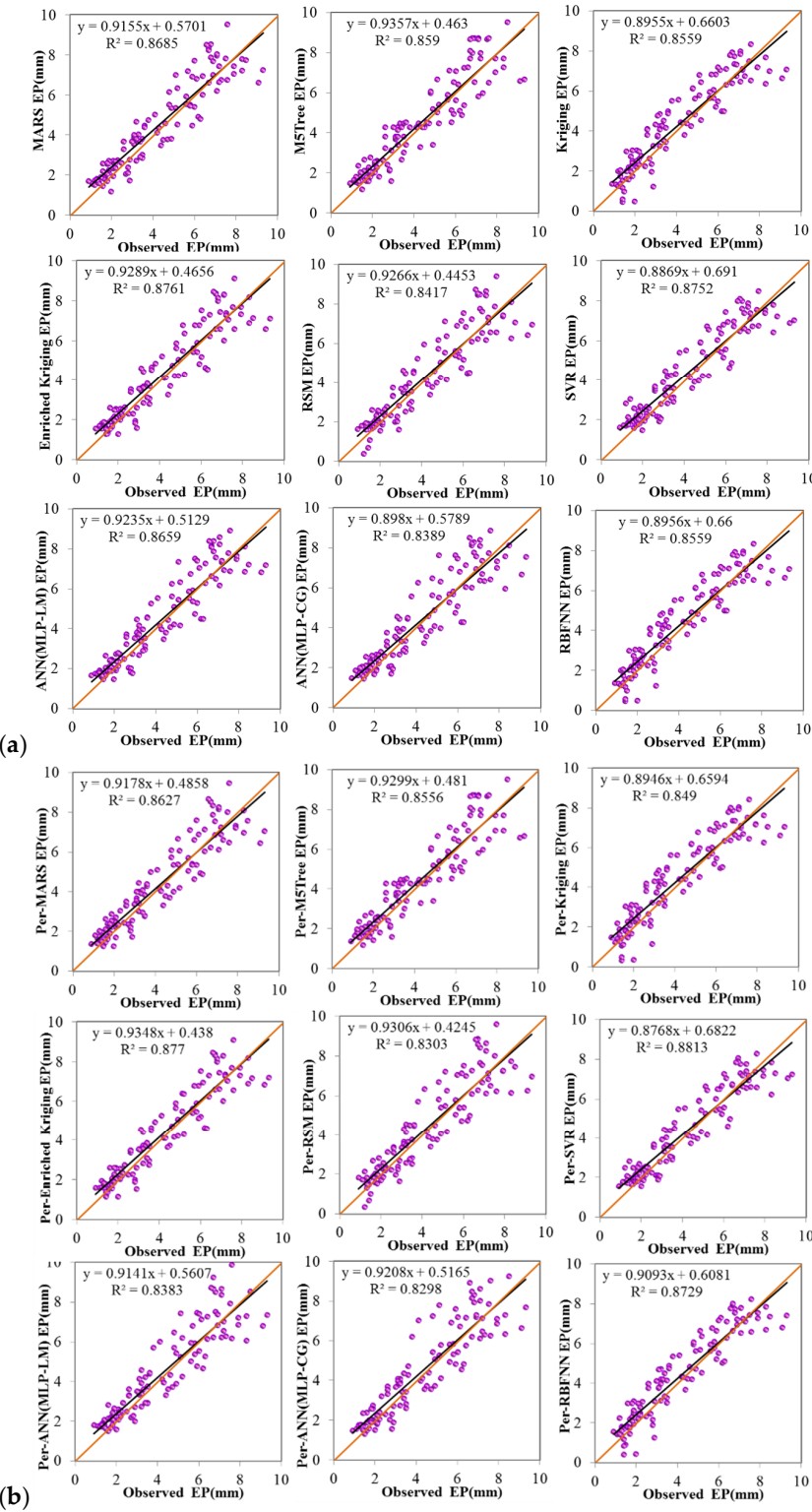

**Figure 6.** The observed and estimated EP (**a**) without and (**b**) with periodicity for Adana station in the testing period.

The comparison statistics of the applied models are given in Tables 4 and 5 for the Antakya Station. In the first scenario, the improved kriging model outperformed the other statistical models considering all of the given measures in Table 4. Nonetheless, in comparison to the SVR—as the best soft computing model—the improved kriging model provided the best performance for the *MBE* (−0.001) value, it failed in sustaining its superiority over the SVR for the *MAE* (improved kriging = 0.489 mm vs. SVR = 0.463 mm) and RMSE (improved kriging = 0.626 mm vs. SVR = 0.613 mm) criteria.

**Table 4.** Comparison of statistical errors for the applied models without periodicity (scenario #1) for Antakya Station in the testing period.

| Category | Model | *MAE* (mm) | *RMSE* (mm) | *MBE* (mm) | *d* | Max (RE) | Mean * (mm) | STD * (mm) | Tot-EP * (mm) | *MAPE* |
|---|---|---|---|---|---|---|---|---|---|---|
| **Statistical** | kriging | 0.555 | 0.717 | −0.190 | 0.974 | 56.81 | 4.34 | 2.17 | 399.39 | 0.142 |
| | Improved kriging | 0.489 | 0.626 | −0.001 | 0.981 | 48.06 | 4.53 | 2.35 | 416.77 | 0.119 |
| | RSM | 0.540 | 0.687 | −0.132 | 0.978 | 60.99 | 4.40 | 2.36 | 404.77 | 0.129 |
| **Soft computing models** | MARS | 0.510 | 0.637 | 0.107 | 0.981 | 60.01 | 4.64 | 2.32 | 426.71 | 0.130 |
| | M5Tree | 0.722 | 0.998 | −0.234 | 0.949 | 58.18 | 4.30 | 2.22 | 395.33 | 0.158 |
| | SVR | 0.463 | 0.613 | −0.012 | 0.981 | 54.12 | 4.52 | 2.18 | 415.77 | 0.115 |
| | MLP-LM | 0.528 | 0.681 | 0.099 | 0.976 | 105.46 | 4.63 | 2.20 | 426.01 | 0.145 |
| | MLP-CG | 0.525 | 0.651 | 0.100 | 0.980 | 49.28 | 4.63 | 2.36 | 426.11 | 0.126 |
| | RBFNN | 0.476 | 0.610 | −0.035 | 0.983 | 47.64 | 4.50 | 2.39 | 413.67 | 0.117 |

* The mean, standard deviation (STD) and total pan evaporation (Tot-EP) of the actual test data points are mean= 4.532 mm, STD = 2.295 mm and Tot-EP = 416.9 mm, respectively. Optimal structure of SVR: (C = 1600, ε = 0.25, σ = 80), ANN(LM): 6-9-1, ANN(CG): 6-7-1, RBFNN: 6-30-1 (σ = 15).

**Table 5.** Comparison of statistical errors for the applied models with periodicity (scenario #2) for Antakya Station in the testing period.

| Category | Model | *MAE* (mm) | *RMSE* (mm) | *MBE* (mm) | *d* | Max (RE) | Mean * (mm) | STD * (mm) | Tot-EP * (mm) | *MAPE* |
|---|---|---|---|---|---|---|---|---|---|---|
| **Statistical** | kriging | 0.560 | 0.721 | −0.188 | 0.973 | 62.26 | 4.55 | 2.35 | 418.14 | 0.145 |
| | Improved kriging | 0.471 | 0.601 | 0.014 | 0.983 | 43.68 | 4.42 | 2.34 | 407.02 | 0.114 |
| | RSM | 0.579 | 0.701 | −0.107 | 0.976 | 67.19 | 4.42 | 2.15 | 406.53 | 0.155 |
| **Soft computing models** | MARS | 0.517 | 0.638 | 0.094 | 0.980 | 48.89 | 4.32 | 2.26 | 397.37 | 0.132 |
| | M5Tree | 0.677 | 0.970 | −0.212 | 0.953 | 49.04 | 4.34 | 2.17 | 399.62 | 0.142 |
| | SVR | 0.496 | 0.664 | −0.113 | 0.977 | 49.49 | 4.53 | 2.24 | 416.47 | 0.117 |
| | MLP-LM | 0.492 | 0.625 | −0.005 | 0.981 | 44.51 | 4.63 | 2.26 | 425.98 | 0.118 |
| | MLP-CG | 0.508 | 0.623 | 0.099 | 0.981 | 68.50 | 4.45 | 2.12 | 409.60 | 0.132 |
| | RBFNN | 0.483 | 0.632 | −0.079 | 0.979 | 48.67 | 4.32 | 2.26 | 397.37 | 0.122 |

* The mean, standard deviation (STD) and total pan evaporation (Tot-EP) of the actual test data points are mean= 4.532 mm, STD = 2.295 mm and Tot-EP = 416.9 mm, respectively. Optimal structure of SVR: (C = 600, ε = 0.3, σ = 80), ANN (LM): 7-12-1, ANN (CG): 7-14-1, RBFNN: 7-30-1 (σ = 15).

For the second scenario (Table 5), the improved kriging surpassed all the other applied statistical and soft computing models considering *MAE*, *MAPE*, *RMSE*, and *d*. In this scenario, the MLP-LM was the best soft computing model in predicting the EP values based on the *MAE*, MBE, *d*, and Max (RE) values. Table 6 presents the best predictive models in terms of three perspectives of accuracy, precision, and tendency. As expected, the improved kriging performed better than the other models in the first scenario (without periodicity), while the MLP-LM and MLP-CG were also among the best models for the second scenario (with periodicity).

**Table 6.** The general performance of the applied predictive models in terms of accuracy, precision, and tendency for Antakya Station in the testing period.

| Category | Model | Scenario I, without Periodicity | | | | Scenario II, with Periodicity | | | |
|---|---|---|---|---|---|---|---|---|---|
| | | Accuracy | Precision | Tendency | Best Model(s) | Accuracy | Precision | Tendency | Best Model(s) |
| Statistical | Kriging | L | H | − | | L | L | − | |
| | Improved kriging | H | H | N | * | H | M | + | * |
| | RSM | L | L | − | | L | M | − | |
| Soft computing models | MARS | M | H | + | | M | H | + | |
| | M5Tree | L | M | − | | L | L | − | |
| | SVR | H | L | − | | M | M | − | |
| | MLP-LM | M | M | + | | H | H | N | * |
| | MLP-CG | M | M | + | | H | H | + | * |
| | RBNN | H | L | − | | M | L | − | |

Note: Accuracy is based on *RMSE* (mm), precision is based on RSTD, and tendency is based on *MBE*. Accuracy and precision: H: high (3 best values), M: moderate (3 median values), L: low (3 worst values); Tendency: −: under-predicted (negative values); +: over-predicted (positive values); N: neutral (absolute value < 0.01 mm). Best models have been chosen based on attaining at least two of the three accuracies (=H), precision (=H), and tendency (=N) criteria.

Figure 7 presents the observed and estimated values of the EP of the Antakya station of applied models a) without periodicity and b) with periodicity. From Figure 7, it is clear that the improved kriging, MARS, and MLP-CG models have similar graphs and they have less scattered predations than the other two models for the two modeling scenarios. It can also be seen that the M5Tree has the most scattered predicted values.

The ratio of the Willmott index of agreement ($d$) to the *MAE* can be used as a measure to compare the accuracy of different models. This statistic ($d/MAE$) varies from 0 to ∞. The larger value of the $d/MAE$ denotes the better calibration of the applied model (Keshtegar et al., 2018). The calculated $d/MAE$ ratios of the applied models are illustrated in Figure 8 for both stations. In general, it is apparent that the improved kriging has higher accuracy than the other models. It can also be observed that better results were given by the improved kriging model considering the periodicity (scenario II). Figure 8 shows that the SVR is the second-best accurate model in predicting EP values, which is similar to the results in Tables 3 and 6 (marked with "H"). Despite being the most accurate soft computing model, the SVR did not act well on the precision and tendency of the predicted values, and as a result, it was not specified as the best model in Tables 3 and 6.

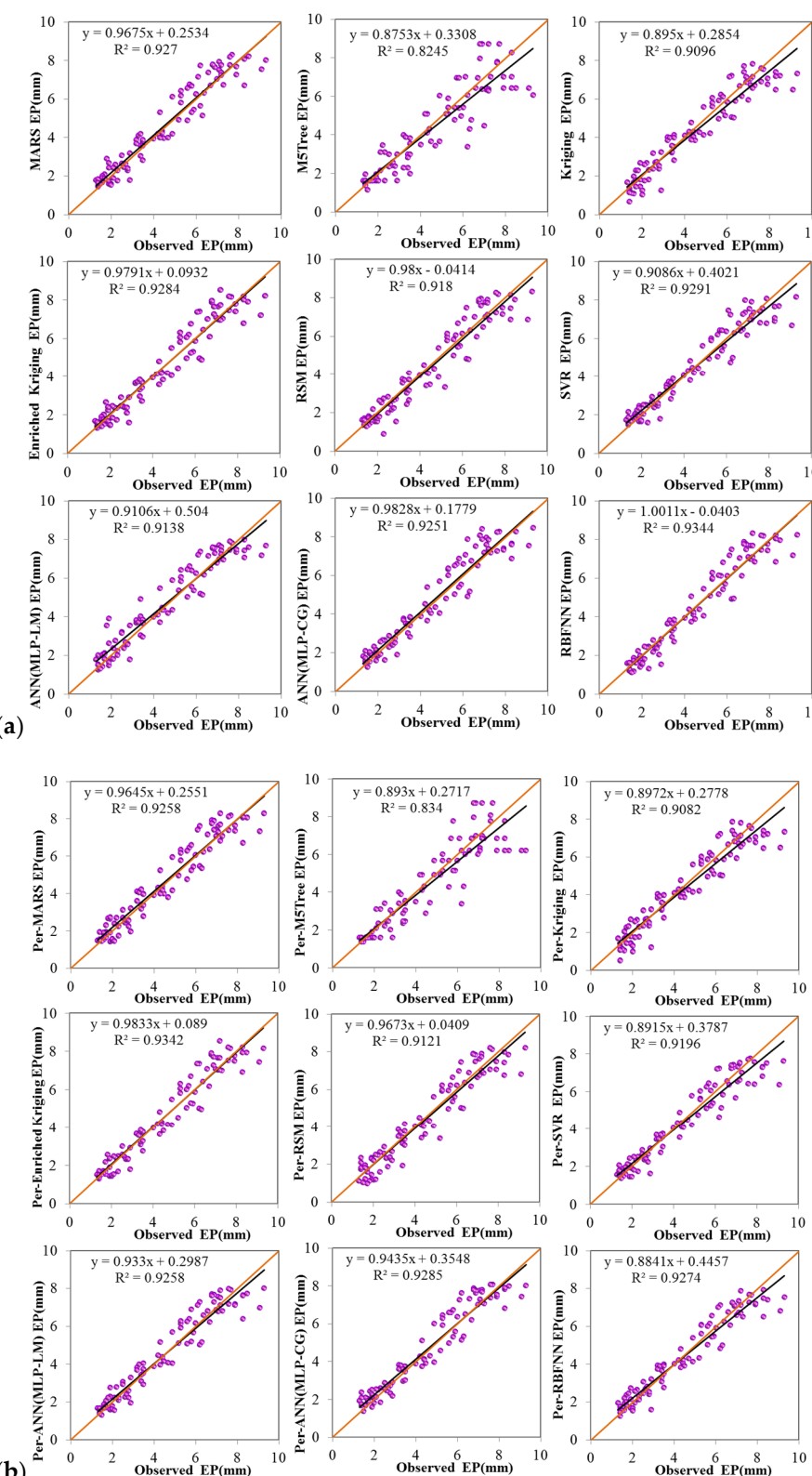

**Figure 7.** The observed and estimated EP (**a**) without and (**b**) with periodicity for Antakya Station in the testing period.

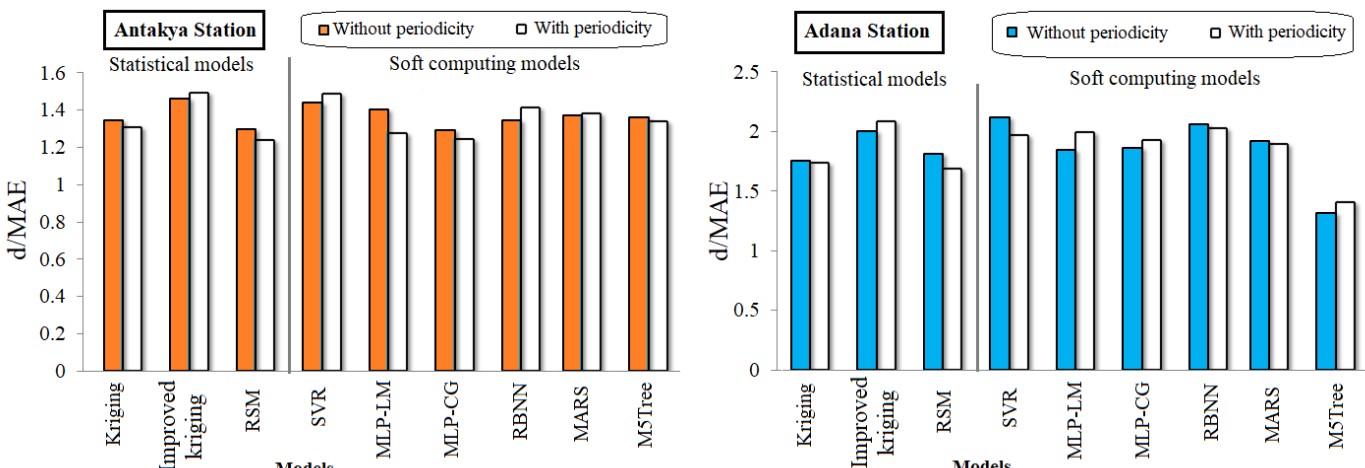

**Figure 8.** Bar charts showing the $d/MAE$ ratio for the applied models in the testing period for Adana and Antakya Stations.

*4.2. Hypothesis Testing*

The results of the significance test of the predicted values of statistical techniques and soft computing models using the Mann–Whitney test are presented in Tables 7 and 8 for the Adana and Antakya Stations, respectively. In the Mann–Whitney test, the null and alternative hypotheses are as follows (η is the median):

**Table 7.** *p*-Values of the Mann–Whitney Test for statistical methods versus soft computing models in Adana Station.

|  |  | Soft Computing Models | | | | | |
| --- | --- | --- | --- | --- | --- | --- | --- |
|  |  | M5Tree | RBNN | MLP-LM | MLP-CG | SVR | MARS |
| **Statistical models** | **RSM** | 0.792 | 0.899 | 0.865 | 0.654 | 0.970 | 0.720 |
|  | **Kriging** | 0.878 | 0.844 | 0.984 | 0.964 | 0.970 | 0.977 |
|  | **Improved kriging** | 0.988 | 0.724 | 0.870 | 0.918 | 0.886 | 0.895 |

**Table 8.** *p*-Values of the Mann–Whitney Test for statistical methods versus soft computing models in Antakya Station.

|  |  | Soft Computing Models | | | | | |
| --- | --- | --- | --- | --- | --- | --- | --- |
|  |  | M5Tree | RBNN | MLP-LM | MLP-CG | SVR | MARS |
| **Statistical models** | **RSM** | 0.824 | 0.873 | 0.626 | 0.631 | 0.786 | 0.638 |
|  | **kriging** | 0.904 | 0.722 | 0.478 | 0.532 | 0.648 | 0.518 |
|  | **Improved kriging** | 0.709 | 0.997 | 0.757 | 0.785 | 0.910 | 0.778 |

The null hypothesis, H0: η1 − η2 = 0.
The alternative hypothesis, H1: η1 − η2 ≠ 0.
The results of Tables 7 and 8 clearly reveal that there is no significant difference between the performance of the statistical models (RSM, kriging, and improved kriging) and soft computing models (M5Tree, RBNN, MLP-LM, MLP-CG, and MARS) at 95% and 99% confidence levels, as they have *p*-values greater than 0.05 and 0.01. In other words, the Mann–Whitney nonparametric test implies that the null hypothesis was not rejected, and none of the applied statistical-based predictive models surpasses the other soft computing models at the 0.05 and 0.01 levels of significance.

## 5. Discussion

This paper aimed to challenge the performance of different statistical and soft computing models based on (i) mathematical (accuracy, precision, and tendency), and (ii) statistical

(at the 0.01 and 0.05 levels of significance) perspectives. In accordance with the mathematical comparisons (Tables 1, 2, 4 and 5, and Figure 8), it was concluded that the improved kriging model performed better than the other applied models, which means that an improved statistical model might even be able to surpass soft computing models.

Figure 9 illustrates the Taylor diagrams for (a) Adana and (b) Antakya Stations. As shown by these figures, the kriging models provide a better prediction for agreement than the RSM but worse than the soft computing models (viz. SVR, MARS, and RBFNN). The SVR provides a superior correlation with the observed data compared to the other soft computing models. As can be seen in Figure 9, the improved kriging enhanced the predictions of the standard kriging model.

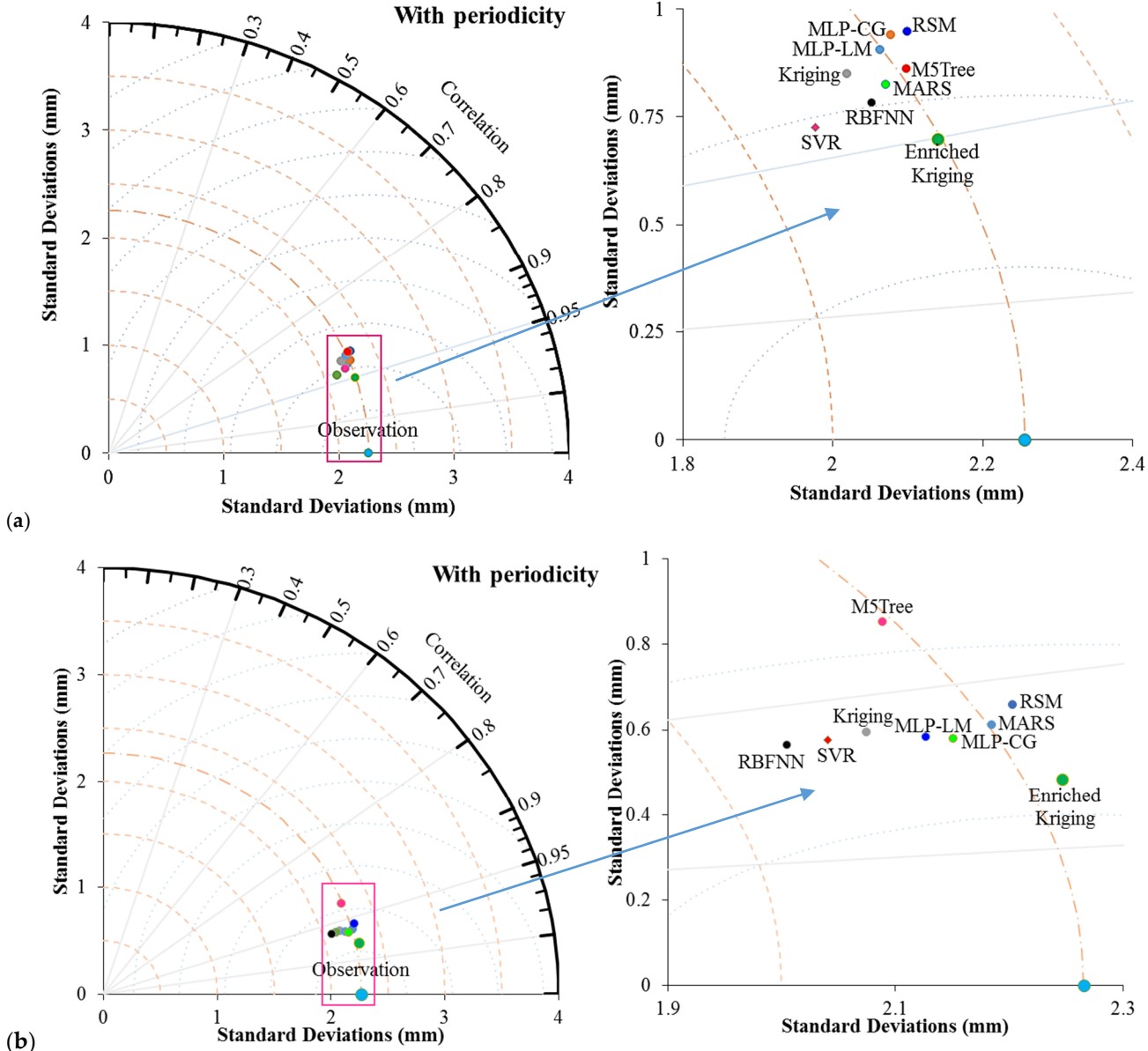

**Figure 9.** Taylor diagram for different models in the testing phase of (**a**) Adana station (**b**) Antakya station.

There is no doubt that in most cases, soft computing models perform better than traditional statistical models. Similarly, the standard kriging and RSM models failed to surpass the soft computing models due to linear cross-correlation regressed function based on the statistical measures. This assessment is based on pertinent studies in the literature. For instance, in a comparative study between the capability of machine learning versus

ANN, and statistical technique versus MLR, for EP prediction, it was found that the model efficiency and correlation coefficient of the ANN was higher than the MLR model for the calibration and validation phases [20]. The same result has been noted for the superiority of ANFIS model over the MLR statistical model [74]. However, it is worth noting that the majority of recent published studies have solely focused on the evaluation of several machine learning models [11,32]. Investigating the outcomes of these studies indicates that the machine learning models perform well in predicting evaporation at different climatical regions.

In this study, in addition to the mathematical evaluation of the potential performance of statistical and soft computing models, the results of the Mann–Whitney hypothesis test were also taken into account. The outcomes of the Mann–Whitney hypothesis test showed that none of the soft computing applied models has significant superiority over the statistical ones. In other words, despite their ability to model nonlinear phenomena, soft computing models should not be taken into granted as the preliminary predictive models. The improved versions of the RSM or kriging-based statistical techniques can improve the accuracy of the prediction of nonlinear problems. Thus, the improved statistical kriging technique uses the exponential transformation of input variables and can also be applied for other engineering problems with nonlinear complex relations. Furthermore, the competency of this method can be apprised by comparing it with machine learning models for complex problems with highly nonlinear relations.

## 6. Conclusions

The soft computing models and statistical techniques are useful frameworks for making predictions of complex climatological indices, such as the hydrological pan evaporation (EP). The improved kriging method was presented as a statistical technique for the accurate prediction of the EP. The RSM, kriging, and improved kriging models were compared with soft computing models, such as the SVR, M5tree, MARS, RBNN, MLP-LM, and MLP-CG. Two different input scenarios, namely with and without periodicity, were applied for the modeling process in the Antakya and the Adana stations located in Turkey. The abilities of statistical models versus soft computing schemes were compared with several statistical measures. The key findings of the study are summarized below:

- Soft computing using machine learning models such as the SVR, MARS, MLP-ML, and RBNN provided more accurate predictions than the M5Tree and RSM.
- The kriging model, as well as the SVR, RBFNN and MLP-ML, provided better performances compared to the RSM and M5Tree.
- It was found that the developed improved kriging model performed better than the other applied models, including the soft computing (SVR, RBNN, MLP-ML, and MARS) and standard statistical (kriging and RSM) models.
- By comparing the performances of the improved kriging method with six other applied models, it can be concluded that the proposed kriging framework can be successfully applied for this current hydrological challenge while its performances for other hydrological stations and other complex, sophisticated problems should be discussed in future.

**Author Contributions:** Conceptualization, M.Z.-K., B.K., O.K. and M.S.; methodology, M.Z.-K. and B.K.; software, B.K. and O.K.; validation, M.Z.-K., B.K. and O.K.; formal analysis, M.Z.-K. and B.K.; investigation, B.K.; resources, O.K.; data curation, O.K.; writing—original draft preparation, M.Z.-K.; B.K. and M.S.; writing—review and editing, M.Z.-K., O.K. and M.S.; supervision, B.K. All authors have read and agreed to the published version of the manuscript.

**Funding:** This work was funded by the University of Zabol with Grant numbers: UOZ-GR-9618-1 and UOZ-GR-9719-1.

**Institutional Review Board Statement:** Not applicable.

**Informed Consent Statement:** Not applicable.

**Data Availability Statement:** Datasets for analysis can give from the co-Author Ozgar Kisi.

**Conflicts of Interest:** We declare that we do not have any commercial or associative interest that represents a conflict of interest in connection with the submitted work.

## Abbreviations

| | |
|---|---|
| BF | Basis functions |
| ANFIS | Adaptive neuro-fuzzy inference systems |
| ANN | Artificial neural networks |
| $d$ | Willmott index |
| ELM | Extreme learning machine |
| LSSVM | Least square support vector machine |
| m | Number basis functions |
| $MAE$ | Mean absolute error |
| $MAPE$ | Mean absolute percentage error |
| MARS | Multivariate adaptive regression spline |
| MBE | Mean bias error |
| MLPNN | Multilayer perceptron artificial neural networks |
| MLR | Multiple linear regression |
| MNLR | Multivariate nonlinear regression |
| $R$ | Correlation matrix |
| RBFNN | Radial basis function neural networks |
| $RMSE$ | Root mean square error |
| SVM | Support vector machine |
| SVR | Support vector regression |
| $w_j$, $w_{ij}$ | Weights |
| $NV$ | Number of input variables |
| $K(x,x_i)$ | Kernel function |
| $\beta$ | Unknown coefficients |

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
