# Peer review of "Towards a Comprehensive Assessment of Statistical versus Soft Computing Models in Hydrology: Application to Monthly Pan Evaporation Prediction"

_water, doi:10.3390/w13172451_

Round 1
Reviewer 1 Report
There is a problem with errors calculation.
MAE is defined as mean average percentage error. However, in tables MAE has mm (!!) as units. So it is not percentage error. Usually MAE is an abbreviation for mean average error - its units are the same as the units of the projected value (mm).
There is MBE defined as mean biase error. However, the formula for MBE looks like the formula for MAPE (mean absolute percentage error) except lack of absolute operator. What is the reason for calculating summed errors where e.g. -0.7 is added to 0.7. The error is 0... falsely. In fact it can be +0.7 or -0.7. Presented MBE is calculated by dividing by the oryginal value, so it is realteive error. But in presented tables it is given in mm (!!??).
There are accuracy and prcission measures are presented in the article. They are measures of classification. It is unclear how this measures are applied for prediction problem.
The result and disscussion can't be assessed due to unclear error calculations.
It is presented in abstract that 9 machine learning models are anaysed. But in the body of the article there are 6 ML models and 3 statistic models.
There are separate tables with the results for "with periodicity" and "without periodicity". The issue of periodicity is not explained.
Histogram presented in fig. 8 is not a histogram.
Author Response
(Q1) There is a problem with errors calculation.
MAE is defined as mean average percentage error. However, in tables MAE has mm (!!) as units. So it is not percentage error. Usually MAE is an abbreviation for mean average error - its units are the same as the units of the projected value (mm).
Response:
There has been a typo in the MAE abbreviation. We have corrected it in the revised manuscript as Mean Absolute Error (MAE), which is a dimensional criterion. It should be noted that all distance measurements (e.g., MAE) have dimensions.
(Q2) There is MBE defined as mean bias error. However, the formula for MBE looks like the formula for MAPE (mean absolute percentage error) except lack of absolute operator. What is the reason for calculating summed errors where e.g. -0.7 is added to 0.7. The error is 0... falsely. In fact it can be +0.7 or -0.7. Presented MBE is calculated by dividing by the oryginal value, so it is realteive error. But in presented tables it is given in mm (!!??).
Response:
We have corrected the written form of the formula for the MBE. We also doubled-checked the results to make sure that they had been calculated based on the authentic MBE equation and realized that the calculated MBE values are correct. Please see Tables 1,2,4, and 6 for your reference.
(Q3) There are accuracy and prcission measures are presented in the article. They are measures of classification. It is unclear how this measures are applied for prediction problem.
Response:
Similar to the simulated values, these measures can also be referred to the predicted values. There are several references that have used these measures for evaluating their results. For example, the following references might be considered:
- Krause, P., Boyle, D. P., & Bäse, F. (2005). Comparison of different efficiency criteria for hydrological model assessment. Advances in geosciences, 5, 89-97.
- Tofallis, C. (2015). A better measure of relative prediction accuracy for model selection and model estimation. Journal of the Operational Research Society, 66(8), 1352-1362.
(Q4) The result and disscussion can't be assessed due to unclear error calculations.
Response:
According to the adjustment in presenting error calculations (Q1 & Q2) as well as explanations to Q4, the evaluation of the model in terms of error calculation has now been clarified.
(Q5) It is presented in abstract that 9 machine learning models are anaysed. But in the body of the article there are 6 ML models and 3 statistic models.
Response:
The mentioned sentence has been corrected.
(Q6) There are separate tables with the results for "with periodicity" and "without periodicity". The issue of periodicity is not explained.
Response:
In “with periodicity” modeling, the time factor has also been considered as an individual independent parameter. This matter has been mentioned in the description of scenarios in section 3.6.
(Q7) Histogram presented in fig. 8 is not a histogram.
Response:
Corrected as bar chart.
Reviewer 2 Report
This paper analyzed and evaluated the competency of statistical and machine learning models in EP modeling. The structure of the paper can be further improved in term of presentation and English, even though, presenting the results is up-to-date and comprehensive. The paper can be considered to be published after revising the following essential issues and drawbacks.
- Why did the authors choose EP for the purpose of their study? where is teh research gap. please elaborate.
- In abstract, and throughout the paper, what does tendency mean?
- In the introduction section, remove the “Null hypothesis: In general, soft computing approaches perform better in modeling hydro-meteorological phenomena than the statistical-based techniques.” Sentence. This is not proper for this type of study.
- Although the introduction section is up-to-date, it is recommended to consider some more recent references to the introduction section from 2020 and 2021.
- it is of interest the way authors showed EP information in Fig 2. However, please provide more explanations regarding the role of boxplots in Fig. 2.
- There are other statistical methods, such as MLR and curve fitting methodologies. Why did you opt the kriging models?
- I can see that the representation of statistics (e.g., MAE and d) is italic in the text, however, these abbreviations are not italic in Tables. Please be consistent.
- Give further information on validation
- State of the art must be improved. Several research in the field had not been included.
- what research had been behind the idea of the presented Improved kriging technique? You should give more explanations.
- The results of the Mann Whitney test were interesting to me. It showed that there is not a significant difference between the applied machine learning models. It seems that the last sentence in the abstract is not valid and should be revised
- Finally, the results of the Mann-Whitney Test showed that the proposed kriging-based model is more accurate than the regular kriging and soft computing models (at α=0.01 and α=0.05).
- move abbrevition to teh end of paper
Author Response
This paper analyzed and evaluated the competency of statistical and machine learning models in EP modeling. The structure of the paper can be further improved in term of presentation and English, even though, presenting the results is up-to-date and comprehensive. The paper can be considered to be published after revising the following essential issues and drawbacks.
(Q1) Why did the authors choose EP for the purpose of their study? where is teh research gap. please elaborate.
Response:
The non-linearity effects of the input can provide a fixable condition for input data in the modelling procedure. By using a kriging-based model, the second order polynomial function, which is the regressed function EP, is used to predict the non-linear challenges. This function may not accurately predict results for complex problems as EP. Thus, we used a non-linear transformation as exponential function for input variables in the EP predictions. The EP can cover these non-linear maps for providing the accurate results in the modelling process of kriging, as presented in the results. However, this modelling method should be examined for other engineering predictions. We have elaborated more on these issues in the methods and discussion parts.
(Q2) In abstract, and throughout the paper, what does tendency mean?
Response:
Tendency refers to the models’ results in terms of over- or under-predication outcomes. In this study, we have used the MBE for introducing this matter. Please see section 3.8.
(Q3) In the introduction section, remove the “Null hypothesis: In general, soft computing approaches perform better in modeling hydro-meteorological phenomena than the statistical-based techniques.” Sentence. This is not proper for this type of study.
Response:
We have changed the text as suggested.
(Q4) Although the introduction section is up-to-date, it is recommended to consider some more recent references to the introduction section from 2020 and 2021.
Response:
We have added some references related to 2020 and 2021 to the literature section of the revised script. Please see the changes in the introduction section.
(Q5) it is of interest the way authors showed EP information in Fig 2. However, please provide more explanations regarding the role of boxplots in Fig. 2.
Response:
The revised text states that these plots graphically depict the variability of each parameter in terms of minimum, quartiles and maximum values. Moreover, outliers are plotted as individual points.
(Q6) There are other statistical methods, such as MLR and curve fitting methodologies. Why did you opt the kriging models?
Response:
The modelling approaches using regression are based on the least square method, and include RSM, kriging, MLR and polynomial chaos expansion. All of these models are regressed by using a polynomial function as linear in MLR, second-order (RMS, Kriging and PCE) or high-order as a modified version of RSM. The kriging model has two terms: (a) a regressing term form the second-order polynomial function; and (b) a stochastic term as kernel error function. The calibration procedure of kriging has a calibrating function and error function for training data points. Thus it is provides an interpolation for training data points with perfect fitness compared to the RSM and PCE. Thus, the non-linearity effects of the predicted data points are improved using the kriging model compared to the RSM and PCE, while it is should be improved for prediction of the complex problem.
(Q7) I can see that the representation of statistics (e.g., MAE and d) is italic in the text, however, these abbreviations are not italic in Tables. Please be consistent.
Response:
Corrected.
(Q8) Give further information on validation
Response:
The time series cross-validation technique has been applied in this study. More information has been added to the text regarding this point. Please see the changes in section 3.8.
(Q9) State of the art must be improved. Several research in the field had not been included.
Response:
We have enriched both the introduction and discussion sections with more related published studies. Please see the changes in these sections.
(Q10) what research had been behind the idea of the presented Improved kriging technique? You should give more explanations.
Response:
This model has been developed based on the authors’ own ideas (see introduction part).
(Q11) The results of the Mann Whitney test were interesting to me. It showed that there is not a significant difference between the applied machine learning models. It seems that the last sentence in the abstract is not valid and should be revised. Finally, the results of the Mann-Whitney Test showed that the proposed kriging-based model is more accurate than the regular kriging and soft computing models (at α=0.01 and α=0.05).
Response:
We have corrected the mentioned sentence in the abstract.
(Q12) move abbrevition to teh end of paper
Response:
We have updated the abbreviation list.
Reviewer 3 Report
The authors have done a good amount of work to justify the estimation of ET with machine learning approach, the amount of work conducted in this work is enormous and further, I believe that there are some problems after addressing those can be considered for publication.
Currently, many of the statements are not supported by published works. Authors may like to find studies in line with their statements to add scientific weight to their observations. I believe that after duly addressing the comments authors can improve the quality of the manuscript substantially to make it more insightful.
Extensive English editing is required as there many problems with sentence restructuring, grammatical errors, punctuations. I suggest authors to consider the English editing a serious concern in this manuscript and with the help of native speaker they can improve this version of the manuscript adequately.
The discussion does not have a proper discussion. There is no citation and comparison with the literature. I recommend them to compare their study with a very recent papers described later in this field
In the introduction, research gaps should be identified better.
I have a big concern in the Introduction, as the authors have missed providing detailed discussion on the important aspect of different classification of ET estimation methods. There is a vast literature on this I would like to suggest few lines following this which author should add is “The ETo estimation models available in the literature may be broadly classified as (1) fully physically-based combination models that account for mass and energy conservation principles; (2) semi-physically based models that deal with either mass or energy conservation; and (3) black-box models based on artificial neural networks, empirical relationships, and fuzzy and genetic algorithms”. I would recommend adding these recent references to add more scientific weight in their Introduction.
Srivastava, A., Sahoo, B., Raghuwanshi, N. S., & Singh, R. (2017). Evaluation of variable-infiltration capacity model and MODIS-terra satellite-derived grid-scale evapotranspiration estimates in a River Basin with Tropical Monsoon-Type climatology. Journal of Irrigation and Drainage Engineering, 143(8), 04017028. https://doi.org/10.1061/(ASCE)IR.1943-4774.0001199
Almorox, J., and GiresserJ., (2016). Calibration of the Hargreaves–Samani method for the calculation of reference evapotranspiration in different Köppen climate classes. Hydrology Research, 47(2): 521-531.
The literature review for the ML techniques is very narrow please try to include some recent studies with the application of ML in ET estimations. Also it would be clear if the authors can describe the novelty of their ML technique than the previous studies and include the study provided here (https://doi.org/10.1007/s00271-018-0583-y).
Line 61-62 authors need to provide the justification of this statement by using the Budyko framework of water-limited ecosystems.
Line correct the symbols of degree at 132 and 134
**Data and Methods**
I would suggest authors provide a flowchart describing the preprocessing of raw datasets obtained from different sources followed by their application.
Also, provide a table detailing all hydrometeorological information with their sources and duration.
Please provide the latitude and longitude of the study area – Figure 1
The clarity of Figure 2 is not clear please improve it.
Figure 3 does not have an axis title.
**Results and Discussion**
Authors are required to add a plot showing the sensitivity of the PM equation to climatic variables. It is important to analyze which variable has a significant effect on ETo estimation for different variables. The amount of change in ETo (mm day−1) with respect to a unit change in each climate variable should be presented in a graphical plot with a daily variation of sensitivity coefficients.
**Conclusion**
Improve the conclusions based on objective and rewrite them in the points.
I would recommend the authors are suggested to provide the future scope of this study either at the end of the Discussion or in the Conclusion section.
Author Response
Reviewer #3
The authors have done a good amount of work to justify the estimation of ET with machine learning approach, the amount of work conducted in this work is enormous and further, I believe that there are some problems after addressing those can be considered for publication. Currently, many of the statements are not supported by published works. Authors may like to find studies in line with their statements to add scientific weight to their observations. I believe that after duly addressing the comments authors can improve the quality of the manuscript substantially to make it more insightful.
(Q1) Extensive English editing is required as there many problems with sentence restructuring, grammatical errors, punctuations. I suggest authors to consider the English editing a serious concern in this manuscript and with the help of native speaker they can improve this version of the manuscript adequately.
Response:
The English of the paper has been checked and revised by a native English speaker.
(Q2) The discussion does not have a proper discussion. There is no citation and comparison with the literature. I recommend them to compare their study with a very recent papers described later in this field
Response:
Done. We have solidified the discussion section by including and comparing the outcomes of other related studies as suggested. Please see the highlighted changes in the discussion section.
(Q3) In the introduction, research gaps should be identified better.
Response:
Research gaps have been identified in the revised paper.
(Q4) I have a big concern in the Introduction, as the authors have missed providing detailed discussion on the important aspect of different classification of ET estimation methods. There is a vast literature on this I would like to suggest few lines following this which author should add is “The ETo estimation models available in the literature may be broadly classified as (1) fully physically-based combination models that account for mass and energy conservation principles; (2) semi-physically based models that deal with either mass or energy conservation; and (3) black-box models based on artificial neural networks, empirical relationships, and fuzzy and genetic algorithms”. I would recommend adding these recent references to add more scientific weight in their Introduction.
Srivastava, A., Sahoo, B., Raghuwanshi, N. S., & Singh, R. (2017). Evaluation of variable-infiltration capacity model and MODIS-terra satellite-derived grid-scale evapotranspiration estimates in a River Basin with Tropical Monsoon-Type climatology. Journal of Irrigation and Drainage Engineering, 143(8), 04017028. https://doi.org/10.1061/(ASCE)IR.1943-4774.0001199
Almorox, J., and GiresserJ., (2016). Calibration of the Hargreaves–Samani method for the calculation of reference evapotranspiration in different Köppen climate classes. Hydrology Research, 47(2): 521-531.
Response:
In the revised version, we have used the above-mentioned references. Also, more recent and updated references have been considered and added to the literature. In addition, we have given more explanations regarding the different attitudes for modeling ETo. Please see the changes in the introduction section for your reference.
(Q5) The literature review for the ML techniques is very narrow please try to include some recent studies with the application of ML in ET estimations. Also it would be clear if the authors can describe the novelty of their ML technique than the previous studies and include the study provided here (https://doi.org/10.1007/s00271-018-0583-y).
Response:
We have considered recent studies along with the mentioned reference for improving the literature review in the revised script. We also provided more information for the novelty of the paper. Please see the added texts related to the novel developed kriging technique at the end of the introduction section.
(Q6) Line 61-62 authors need to provide the justification of this statement by using the Budyko framework of water-limited ecosystems.
Response:
Please see the modified first paragraph of the introduction section.
(Q7) Line correct the symbols of degree at 132 and 134
Response:
Corrected.
**Data and Methods**
(Q8) I would suggest authors provide a flowchart describing the preprocessing of raw datasets obtained from different sources followed by their application.
Also, provide a table detailing all hydrometeorological information with their sources and duration.
Response:
We have added more information to the case study and data sections.
(Q9) Please provide the latitude and longitude of the study area – Figure 1
Response:
We have now mentioned the latitude and longitude of the two stations in the caption of Figure 1.
(Q10) The clarity of Figure 2 is not clear please improve it.
Response:
Done.
(Q11) Figure 3 does not have an axis title.
Response:
Corrected.
**Results and Discussion**
(Q12) Authors are required to add a plot showing the sensitivity of the PM equation to climatic variables. It is important to analyze which variable has a significant effect on ETo estimation for different variables. The amount of change in ETo (mm day−1) with respect to a unit change in each climate variable should be presented in a graphical plot with a daily variation of sensitivity coefficients.
Response:
We have executed and added the results of the sensitivity analysis to section 2.
**Conclusion**
(Q13) Improve the conclusions based on objective and rewrite them in the points.
I would recommend the authors are suggested to provide the future scope of this study either at the end of the Discussion or in the Conclusion section.
Response:
Please see the changes in the discussion and conclusion sections.
Round 2
Reviewer 1 Report
There is still the problem in errors calculations.
- According to formula (18), as average, all models give overestimated values. None of them give negative MBE. What suggests, that the models are not worked out (their training procedures) or relative errors MAPE are very high as MBE (averaged sum of residuals without applying absolute operator in nominator of formula 18) is close to 10 %.
- Max(RE) is from 2.28 to 2.85 (the formula for maxRE is presented in line 371). It means that if the observed PE is 1.2 mm, the predicted value can be from 0 to 3 mm. If the observed value is 9 mm, the predicted one can be from 0 to 26 mm. That is the reason why MAPE (mean absolute percentage error) is widely applied (but not in this article).
- Rel-Error (presented in the last column, in tab 1 and tab 2) is not defined.
- Presenting mean predicted values and referring it to mean observed values has no meaning. For the following to pairs (observed, predicted): (10, 5) and (10, 15) means are the same (10) but relative errors are huge (50 %).
In my opinion error calculations presented in the article are still inconsistent and should be worked out.
Author Response
- According to formula (18), as average, all models give overestimated values. None of them give negative MBE. What suggests, that the models are not worked out (their training procedures) or relative errors MAPE are very high as MBE (averaged sum of residuals without applying absolute operator in nominator of formula 18) is close to 10 %.
Response:
The authors checked the formula and results and agree with the reviewer. As you can see in the revised script, the correct MBE formula has now been provided. Also, there have been some changes for the updated MBE values. However, for the first scenario, all models gave positive values for MBE, which is not the case for the second scenario. This outcome has been shown and reflected in Tables 3 and 6, where in Table 3, the tendency for the applied models is positive (over-predicted). Nonetheless, Table 6 indicates that some of the models have tendencies to under-predict and others to over-predict.
We also calculated the MAPE for comparing the results using Eq. (21) in the revised manuscript. All the calculated values presented in Tables 1, 2, 4 and 5 have been recomputed.
- Max(RE) is from 2.28 to 2.85 (the formula for maxRE is presented in line 371). It means that if the observed PE is 1.2 mm, the predicted value can be from 0 to 3 mm. If the observed value is 9 mm, the predicted one can be from 0 to 26 mm. That is the reason why MAPE (mean absolute percentage error) is widely applied (but not in this article).
Response:
We have corrected the Max(RE) values in the revised manuscript as highlighted in Tables 1, 2, 4 and 5. The new Max(RE) values are now in the appropriate range.
- Rel-Error (presented in the last column, in tab 1 and tab 2) is not defined.
Response:
This issue has been resolved by removing Rel-Error, which is equal to the bias error BE. Since we have reported on MBE, Rel-Error does not give any extra information.
- Presenting mean predicted values and referring it to mean observed values has no meaning. For the following to pairs (observed, predicted): (10, 5) and (10, 15) means are the same (10) but relative errors are huge (50 %).
Response:
Based on the corrected values for Max(RE), it can now be seen that the relative errors are high and more than 50%, as indicated by the reviewer. Please refer to Tables 1, 2, 4 and 5.
In my opinion error calculations presented in the article are still inconsistent and should be worked out.
Response:
We have corrected the mathematical measures and upgraded the results accordingly.
Reviewer 3 Report
I am happy to say that the authors have significantly improved the manuscript and in that process, it is now in publishable form. However just a few minor concerns:
Include some key results in the abstract section
Figure 1 Improve the clarity of the figure and provide the latitude and longitude
Also, improve the structuring of paragraphs in the conclusion as the last few paragraphs can be merged.
Author Response
I am happy to say that the authors have significantly improved the manuscript and in that process, it is now in publishable form. However just a few minor concerns:
Include some key results in the abstract section
Response:
We have reinforced the findings by presenting more comparisons along with mathematical results in the abstract. Please refer to our changes highlighted in green.
Figure 1 Improve the clarity of the figure and provide the latitude and longitude
Response:
Figure 1 has now graphical indications for latitude and longitude.
Also, improve the structuring of paragraphs in the conclusion as the last few paragraphs can be merged.
Response:
The key conclusions are now presented using bullet points.
Round 3
Reviewer 1 Report
According to formula 21 and the values of MAPE presented in Table 2, if the predicted value of PE is 1 mm, as average (based on MAPE 20.0) the real value can be from 0 to 20 mm (what is even out of the range of the original values). If the prediction is 5 mm, we can expect, as average, EP ranging from 2 to 100mm (for MAPE 20.0).
I suspect a serious flaw in the calculations. If MAPE presented is true, so huge error (MAPE) disqualify the utility of such predictions.
Author Response
According to formula 21 and the values of MAPE presented in Table 2, if the predicted value of PE is 1 mm, as average (based on MAPE 20.0) the real value can be from 0 to 20 mm (what is even out of the range of the original values). If the prediction is 5 mm, we can expect, as average, EP ranging from 2 to 100mm (for MAPE 20.0).
I suspect a serious flaw in the calculations. If MAPE presented is true, so huge error (MAPE) disqualify the utility of such predictions.
Response
Thank you for your kind observation, author recheck and corrected them, it is a program mystic that it is improved and results are corrected in revised manuscript.